# Personal VOCs Exposure with a Sensor Network Based on Low-Cost Gas Sensor, and Machine Learning Enabled Indoor Localization

**DOI:** 10.3390/s23052457

**Published:** 2023-02-23

**Authors:** Leonardo Papale, Alexandro Catini, Rosamaria Capuano, Valerio Allegra, Eugenio Martinelli, Massimo Palmacci, Giovanna Tranfo, Corrado Di Natale

**Affiliations:** 1Department of Electronic Engineering, University of Rome Tor Vergata, Via del Politecnico 1, 00133 Rome, Italy; 2Department of Occupational and Environmental Medicine, Epidemiology, and Hygiene, Istituto Nazionale Assicurazione Infortuni sul Lavoro, Monte Porzio Catone, 00144 Rome, Italy

**Keywords:** Wireless Sensor Network, indoor localization, volatile organic compounds

## Abstract

Indoor locations with limited air exchange can easily be contaminated by harmful volatile compounds. Thus, is of great interest to monitor the distribution of chemicals indoors to reduce associated risks. To this end, we introduce a monitoring system based on a Machine Learning approach that processes the information delivered by a low-cost wearable VOC sensor incorporated in a Wireless Sensor Network (WSN). The WSN includes fixed anchor nodes necessary for the localization of mobile devices. The localization of mobile sensor units is the main challenge for indoor applications. Yes. The localization of mobile devices was performed by analyzing the *RSSIs* with machine learning algorithms aimed at localizing the emitting source in a predefined map. Tests performed on a 120 m^2^ meandered indoor location showed a localization accuracy greater than 99%. The WSN, equipped with a commercial metal oxide semiconductor gas sensor, was used to map the distribution of ethanol from a point-like source. The sensor signal correlated with the actual ethanol concentration as measured by a PhotoIonization Detector (PID), demonstrating the simultaneous detection and localization of the VOC source.

## 1. Introduction

Indoor locations, sometimes characterized by a limited exchange of air, are exposed to the accumulation of chemicals whose concentration may easily grow beyond the allowed limits. This problem is common in both domestic and recreational environments, but in some working places it may become particularly important. For example, around 60% of workers in agriculture and 50% in the manufacturing and construction fields are exposed to noxious substances [1].

Considering that the effect of exposure depends on the nature of the chemical and on the exposure time, it is essential to timely control the dispersion of chemicals in the air and the dose at which operators are exposed. This scope can be achieved by a personalized monitoring system that measures the concentration of chemicals in the air immediately surrounding an individual. In this context, Wireless Sensor Networks (WSNs [2]) are an optimal solution for implementing monitoring systems [3,4].

A complete WSN is composed of mobiles and fixed nodes. The mobile nodes are designed to detect specific chemicals, analyze the data, identify the presence of harmful substances, and communicate the information to the central node. To obtain information about the inhaled substances, the sensing elements should be wearable, with consequent constraints in size and weight. The central node could be connected to a Personal Computer (PC), characterized by suitable computational capability, large storage, and an internet connection.

A necessary functionality of the system is the localization of the mobile units inside the monitored area. This function facilitates the localization of contaminated or dangerous areas to ensure immediate countermeasures. Furthermore, a suitable integration of the collected signals can turn the mobile device into a dosimeter for personal long-term total exposure evaluation.

Accurate indoor localization can be achieved with different technologies. Most of the investigated solutions have been based on RF signals, but other technologies, such as ultrasound (Active BAT [5], Cricket [6]), visible light [7], and infrared [8], have also been proposed. Among them, RF appears to be more competitive; for instance, RF signals can penetrate obstacles, making localization possible even in non-line-of-sight conditions. Different properties of the RF signal can be used to evaluate the distance between the transmitter and the receiver. The main ones are based on the power of communication, which is generally expressed as Received Signal Strength Indication (*RSSI*), Time of Flight (ToF), and Angle of Arrival (AoA) [9]. Among them, *RSSI* does not require specific hardware or synchronization capabilities. In addition, the *RSSI* parameter is included in most wireless communication technologies. As a drawback, the received power is a noisy quantity, so it needs to be processed to extract the relevant information for optimal position estimation. The application of *RSSI* in a WSN requires a number of fixed devices called anchor nodes that act as a re-transmitter between mobile and central nodes. The anchor nodes measure the *RSSI* of incoming communications and send the value to the central node for localization.

The relationship between *RSSIs* and the node’s location can be determined by many approaches, generally divided into two categories: range-based [10] and range-free [11]. Range-based techniques use specific signal propagation models to estimate the distance from *RSSI* values. The main advantage is that these solutions do not require a calibration step. Additionally, theoretical model parameters often do not work correctly for the environment considered; in this case, it is possible to perform measurements to obtain the most suitable values [12]. Recently, machine learning has been combined with range-based strategies, leading to an improvement in localization performance [13]. Instead of using *RSSI* to evaluate a distance, the range-free techniques associate positions with *RSSI* patterns. Although they require an initial calibration step that usually takes only a few minutes, they generally give better results because they can adapt to the environment where they are applied. Specifically, we adopted scene analysis, a machine learning-based range-free approach [14].

Performance is partially determined by communication technology. Usually, the best localization results are obtained with Wi-Fi signals. In this context, deep learning neural networks showed interesting results in terms of position error minimization [15,16]. However, Wi-Fi signals may be unfavorable because of excessive power consumption. In this regard, Bluetooth Low Energy (BLE) provides the best trade-off between the accuracy of localization and the power consumption [17,18,19,20].

In this paper, a WSN made of five mobile sensing nodes, three anchor nodes, and a central node is illustrated. The WSN implements integrated gas sensor technology, BLE communication, and machine learning algorithms. The WSN was aimed at monitoring VOCs in an indoor area. The simplified system scheme is depicted in Figure 1 where the communication paths from the mobile sensing node, the anchor node, and the central node are shown.

Mobile sensing units were equipped with an integrated metal oxide semiconductor gas sensor. The capability to track the distribution of volatile compounds was tested in a simulated experiment aimed at detecting the distribution of ethanol released by a point-like continuous source. The sensor signals were compared with those of a Photoionizing Detector (PID) placed in the same position as the sensing node. Sensor signals correlated, as expected, with the concentration of ethanol, demonstrating the capability of the sensor to detect ethanol at ppm concentration level in free air.

## 2. Materials and Methods

### 2.1. WSN

The Wireless Sensor Network comprises three kinds of nodes each with a unique and specific role: sensing, anchor, and central nodes. The sensing nodes are portable devices whose main task is to detect environmental variables. Sensing nodes communicate the collected data to the anchor nodes through a BLE connection. Anchor nodes receive the data, measure the *RSSI* of the communication, and deliver this information to the central unit using the ESPNOW protocol. The information path is shown in Figure 2. Sensing nodes are composed of an ESP32 microcontroller, a 2000 mAh LiPo battery that guarantees autonomy to the device, and a sensor array. Anchor and central nodes have an ESP32 microcontroller and are directly powered by the electrical grid. The ESP32 was mounted on the development board Lolin32 to ensure low cost and low power consumption. ESP32 supports both BLE and Wi-Fi technology.

The sensor array was designed to be customizable with gas sensors of different kinds using I2C serial communication and ADC analog inputs. Here, the BME680 sensor from Bosch was used [21]. This device includes a metal oxide semiconductor (MOx) gas sensor complemented by temperature, humidity, and pressure sensors. MOx sensors are highly sensitive and broadly selective to ensure a reliable evaluation of total VOCs concentration. The relevant data are the resistances of the MOx kept at a specific working temperature. The node, composed of a microcontroller, battery, and sensor, is packaged in a 3D printed case ready to be worn as an armband kept in place by a Velcro strap.

The sketch of the sensing node is depicted in Figure 3. The total weight is 75 g, and the dimension is 6 × 5 × 3.5 cm. The sensing node is sampled once every second. The node broadcasts the signals of the sensor array during communication. The classical BLE advertising process uses three channels, called 37, 38, and 39. It is known that with these channels, the localization accuracy improves [22,23]. However, since the BLE protocol does not provide the channel information, it is almost impossible to correctly understand the *RSSI* readings, which change remarkably from channel to channel. A single channel was used to overcome this problem and obtain a stable *RSSI* signal.

As communication parameters, we used an advertising interval of 12.5 ms and a communication period of 200 ms [24]. In such a way, the communication will be highly redundant with a manifold of *RSSI* values in a short time. The abundance of signals from the same location reduces the loss of information and facilitates localization. The anchor node constantly scans the communication channel to receive the BLE communication from the sensing node. The receiver module of the ESP32 microcontroller receives the communication and measures the related *RSSI*. Furthermore, it demodulates the signal and converts it to the digital domain using embedded high-speed analog-to-digital converters (ADCs). In case of multiple receptions of the same packet, the anchor node stores the *RSSI* readings for localization.

The data measured from sensors and the *RSSI* readings are rearranged in the ESPNOW payload and sent to the central node. ESPNOW is a point-to-point, Wi-Fi-based protocol of communication [25]. Even if the power consumption is not as low as that of BLE, returning bigger payloads and longer distances is still preferable. In this way, most of the packet’s loss will not happen in the second communication step, but rather in the first one. The central node is connected to a PC via the USB port to transfer the information received using serial communication. The PC runs a custom MATLAB script whose GUI is shown in Figure 4. It processes the data and applies the classification model to the *RSSI* data. The classification model is trained to return the location according to the areas evidenced in the map. Different modalities of data presentation can be applied. For instance, sensor data can be compared with pre-defined thresholds and the position of the sensing node that detected the excess of VOC can be shown. Additional values include the data of sensors and the level of battery of the nodes.

### 2.2. Indoor Positioning

The scene analysis is a typical *RSSI*-based approach, it consists of collecting data from the different locations inside an area. The data are then used to create a model that can estimate the positions of the sensing nodes. The monitored area has been divided into zones of limited dimensions. The number and the size of these zones are arbitrary; they can be determined from the expected diffusion of the chemicals, the occupancy, and the use of the spaces. The deployment of the anchor nodes is crucial, and it can be optimized, for instance, with repeated trials. To calibrate the model, a certain number of *RSSIs* were collected from each area. The calibration dataset was made by 120 spatially distributed measurements for each zone. Each measurement was described by a pattern of three features corresponding to the *RSSI* values measured by the three anchor nodes. The calibration data were used to train the classification model. The trained model was then used by the MATLAB script to locate on the map the signals from the mobile nodes.

### 2.3. RSSI Features

The sensing node advertises on channel 37 multiple times the same packet at an interval of 200 ms. In this interval of time, the anchor node receives multiple *RSSI* readings associated with the same location. Indeed, it is plausible to assume that, under any condition, in 200 ms the mobile node does not change position. The number of *RSSI* readings is variable because of several factors, such as interferences, distances, and BLE random delays. The n
*RSSI* readings associated with a single packet are assembled into a representative indicator using one of the following three possibilities: the maximum, the average, or the maximum ratio combining (MRC).

The choice of the maximum value considers the fact that the *RSSI* decreases due to indoor fading. Under this circumstance, the maximum value among those available could be the most representative indicator. It is computed as follows:(1)$$RSSI_{Max}=max\left(RSSI_{1},\;\ldots \;RSSI_{n}\right)$$

In the case of the noise components of the radio frequency signals, the mean value is more robust with respect to the maximum value. It is computed as follows:(2)$$RSSI_{Mean}=\frac{1}{n}\cdot \sum_{i=1}^{n}RSSI_{i}\;$$

The maximum ratio combining (MRC) weights the *RSSI* reading according to its value. Larger *RSSIs* therefore contribute more than those of smaller value. It is computed as follows:(3)$$RSSI_{Mrc}=\sum_{i=1}^{n}RSSI_{i}\cdot w_{i}\;$$
where:(4)$$w_{i}=\frac{RSSI_{i}-RSSI_{MIN}}{\sum_{j=1}^{n}RSSI_{j}-RSSI_{MIN}}\;$$
where *RSSI_MIN_* is a reference value. Here, it has been fixed at *RSSI_MIN_* = −90 dBm, a value close to the Receiver Sensitivity of the ESP32 communication module. The Receiver Sensitivity is the minimum signal strength that a receiver can detect.

The efficiency of each indicator to provide an accurate localization is discussed later.

### 2.4. Filtering

The *RSSI* features calculated with the above equations are affected by several disturbances, including signal attenuation due to transmission through obstacles, and fluctuations due to multipath fading and indoor noise [26]. Filtering is essential to clean the signals and enhance the performance of the localization algorithms. This problem is well known, and filtering techniques of *RSSI* are reported in the literature. Here, they are applied to the BLE signals. Since sensing nodes transmit a new packet every second, filtering algorithms can operate on sequences of values collected in a specific time window. Regardless of the chosen algorithm, the filter introduces an initial delay in the order of the adopted time window. After the initial delay, a new value is calculated each second. In addition, the same delay occurs any time the mobile units move from one zone to another. To limit the delay time, the time window must be chosen as narrow as possible. These considerations hold for any filtering algorithm.

Of the available filtering approaches, we applied moving average, median filter, and maximum filter. All filtering algorithms depend on the width of the time window, hereafter indicated as m. The time window is given in seconds, and it is always an integer multiple of one second.

Moving average evaluates the average of a sequence of *m* collected values, and it replaces the central value of the sequence with:(5)$$F_{Mean,i}=\frac{1}{m}\cdot \sum_{j=1-m/2}^{j=1+m/2}(F_{j})$$

The median filter aggregates the sequence of *m* values and replaces the central value of the sequence with the median value:(6)$$F_{Med,i}=median\left(F_{j}\right)\;\;with\;j=1-m/2;\;..\;1+m/2\;$$

The maximum filter evaluates the maximum in the sequence of *m* values and replaces the central value of the sequence with:(7)$$F_{Max,i}=max\left(F_{j}\right)\;\;\;\;\;\;with\;j=1-m/2;\;..\;1+m/2$$

The performance of the filters is evaluated by comparing the accuracy of localization and the magnitude of the initial delay.

### 2.5. Classification Models

The classification models were calculated on the calibration dataset. This is composed of 120 elements for each area of the map. Each area of the map is considered a class of the classification problem. The localization with machine learning has been attempted in the past using different algorithms such as k-Nearest Neighbor (k-NN, [20]), weighted k-NN algorithms ([27,28]), and Artificial Neural Networks (ANNs, [16]).

Here, k-NN, Weighted k-NN [20], Decision Tree, Linear Discriminant Analysis, and Gaussian Naive Bayes classifiers were applied and compared.

In particular, the Artificial Neural Network was implemented using one hidden layer with ten neurons and Rectified Linear Unit (ReLU) activation functions. For the output layer, a SoftMax activation function was used.

Through an optimization process, we selected the optimal values of k for the k-NN and weighted k-NN algorithms. We consistently obtained better classification performances with low values of k, therefore we used k equal to five for most of the models.

Weighted k-NN is a powerful alternative to the classical k-NN algorithm. Given an input element, it associates a weight, inversely proportional to the distance, to each element of the k set of neighbors. Among the available definition of weights, we chose the square of the distance (*d_i_*) of the k-th neighbor from the input:(8)$$w_{i}=\frac{1}{d_{i}{}^{2}}\;$$

All classifiers were calculated with 70% of the dataset as training data and 30% as test data. The classification accuracies, defined as the percentage of correctly identified signals, were compared to evaluate the classifier’s performance. The training was repeated ten times for each classifier to obtain statistically robust results. The mean and variance of classification accuracies on the test set were considered.

### 2.6. Experimental Setup

Tests were performed on the floor of a building composed of offices and laboratories. The floor map is shown in Figure 5 We used one of the corridors, approximately 25 m long, and several rooms. These spaces are typically occupied in both stationary and dynamic ways, generally by one or two stationary people in each room with people crossing the corridor. The calibration step was performed using a representative dataset that also considered the presence of humans and dynamic obstacles. The chosen zones were the classes of the classification algorithm. Using more zones enhances the system’s spatial resolution but can deteriorate classification performances since the system will mislead them more frequently. In addition, to choose the dimension of the zones, we must consider the system’s sensing performance bonded to the gas’s diffusion process. We walked in each area, simulating normal human behavior, acquiring 120 elements to obtain a representative dataset. In the first localization experiment, we considered a region with a limited space of about 50 m^2^. We divided the space into seven zones, as shown in Figure 5, to test the different solutions proposed and optimize the entire system in this way. The squared zones were approximately 2.5 × 2.5 m^2^, while the corridor was divided into square areas of roughly 3 × 1.5 m^2^. The anchor node position that gives optimal results depends strongly on the environment properties and should be identified through an optimization process. In the first localization experiment, we distributed the anchor nodes in positions that reasonably help discriminate between the seven areas considered. In the second experiment, the dimension of the covered area was extended due to the optimization of the location of the anchor nodes. Finally, the total area of 120 m^2^ was divided into 16 zones, as shown in Figure 6.

A VOC leakage accident was simulated to evaluate the system’s sensing performance. A point-like continuous source was simulated by a glass beaker, with a surface of 7 cm^2^, filled with 50 mL of ethanol. Ethanol did not completely evaporate during the test, so it provided a continuous and constant source of volatile molecules. The actual concentration of ethanol was measured by a Photoionization Detector (Model: MiniPID2 PPM by ION Science [29]). The mobile node was moved to different locations of the test site (see Figure 5) and kept stable until a steady resistance value was achieved.

## 3. Results and Discussion

### 3.1. Localization Experiments

A first run of localization measurements was aimed at optimizing the parameters of the classifiers. Here, the results achieved by the optimal combination of parameters are shown. To determine which feature achieved the best performance, we compared the classification accuracy of the filters using the maximum, the mean, and the MRC for each combination of classifiers and filters. Before applying the different classifiers, we investigated the intrinsic discrimination capability of the dataset using Principal Component Analysis (PCA). The scores plot of the first two principal components is shown in Figure 7. The distribution of the data shows the discrimination capability of the features and anticipates the classification results achieved by the supervised algorithms. The results indicate that the maximum value is the feature that gives better classification accuracies, independently of both the classifiers and the filters. As an example, Figure 8 shows the classification accuracy of classifiers trained with the three features and a moving average filter. Furthermore, the MRC generally works better than the mean value because it is usually closer to the maximum. These results confirm that the maximum value includes more information about the position than the other features. Once the best feature has been selected, we can study the impact of filters.

As shown in Figure 9, applying moving filters significantly improved the performance. The moving average filter gave better results for any classifier because the classification accuracy curves increased faster, smoother, and monotonically. Finally, to evaluate the best classifier, we must consider both the classification accuracy and the time delay introduced by the filter. For example, we set a threshold of 99% of correct classification and measured how many seconds the classifier crosses to guarantee the best performance with limited delay. Applying the moving average, as shown in Figure 9a, four classifiers stand out in terms of performance: k-NN, weighted k-NN, ANN, and Naïve Bayes. Among these, the weighted k-NN performed better, crossing the threshold in only six seconds, a time delay usually acceptable for most applications.

In a second test, the anchor nodes were distributed along the corridor to cover a larger area, as shown in Figure 6. The results are depicted in Figure 10. The classification problem was more challenging than in the previous experiment because of the increased number of areas and the wider region covered. The growing complexity is easily visible from the results. Even if the moving average filter always has a positive effect, we need a more significant time window to achieve acceptable performances. The best results were obtained with the weighted k-NN, which crossed the 99% threshold with a time window of 10 s.

The results show that the position of the anchor node is crucial, and it dramatically affected the classification performance. The configuration used, where the anchor nodes were distributed along the corridor, is optimal for communication purposes. It defines an axis of spatial symmetry that can lead to classification problems because the areas mirrored concerning that axis will give similar *RSSI* patterns. This is observed when we add two other zones, 17 and 18, as shown in Figure 11. The effect of this addition is a performance drop, since zone 17 will be confused with 8 and 9, and zone 18 with 1, 2, and 3. To overcome this issue and cover a wider region, we should add other anchor nodes that break the spatial symmetry.

### 3.2. VOC Leakage Experiment

The VOC leakage experiment simulated the accidental release of chemicals into the air. Tests were performed trying to approximate a still-air condition, avoiding turbulence and drift. Measurements were performed with an approximately constant distribution of ethanol. The actual concentration was measured by PID. Figure 12a shows the PID signals in various locations. In the figure, the places where measurements were taken are highlighted. The concentration of ethanol ranged from 61 ppm (close to the source) to 3.23 ppm at the farthest position.

The MOx sensor contained in the BME680 is a SnO_2_ semiconductor. It is known that in these sensors the interaction with reducing gases, such as ethanol, increases the conductivity and decreases the resistance. The relative variation of resistance with respect to a baseline value (*R*_0_) measured in absence of ethanol is expressed as $\frac{R_{s}}{R_{0}}$.

Figure 12b shows the relative change in the resistance versus the ethanol concentration. The data are fitted by a typical power law relationship with power −0.45, which is very close to the theoretical −0.5 [30].

This last result highlights the potential of the system and indicates the possible applications of the WSN.

## 4. Conclusions

This paper illustrates a WSN endowed with a gas sensor and indoor localization. The choice of technology and the design aimed at a low-cost and low-power sensing node capable of rapidly revealing dangerous situations and communicating these events to a central unit. *RSSI* has been used as a feature to detect the position of the mobile units. The location was enabled by a machine learning algorithm that processed the *RSSI* data.

Localization accuracy was optimized by selecting a combination of features, filters, and classifiers. We demonstrated that this system, with three anchor nodes, can cover an area of 120 $m^{2}$ made of individual rooms and corridors with a localization accuracy larger than 99%.

A VOC leakage accident was simulated to evaluate the sensing performance by a point-like and constant release of ethanol vapors. The results show the correlation between sensor signals and the actual concentration measured by a PID.

The system can be promptly expanded by increasing the number of anchor nodes to cover larger areas and obtain better spatial resolution. Finally, after a proper sensor calibration, the system could be ready to be used in real situations to monitor and rapidly detect exposure to specific volatile compounds.

## Figures and Tables

**Figure 1 sensors-23-02457-f001:**
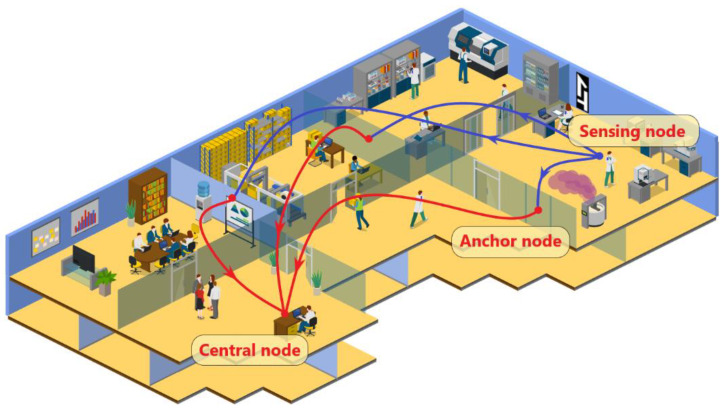
Representation of the WSN proposed, with the localization functionality, applied in a generic indoor environment.

**Figure 2 sensors-23-02457-f002:**
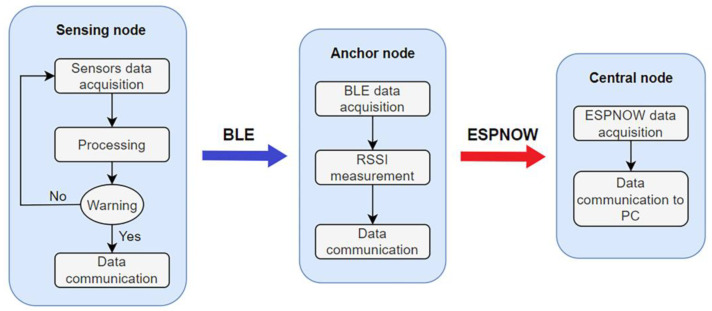
Block diagram of sensing, anchor, and central nodes and their communication path.

**Figure 3 sensors-23-02457-f003:**
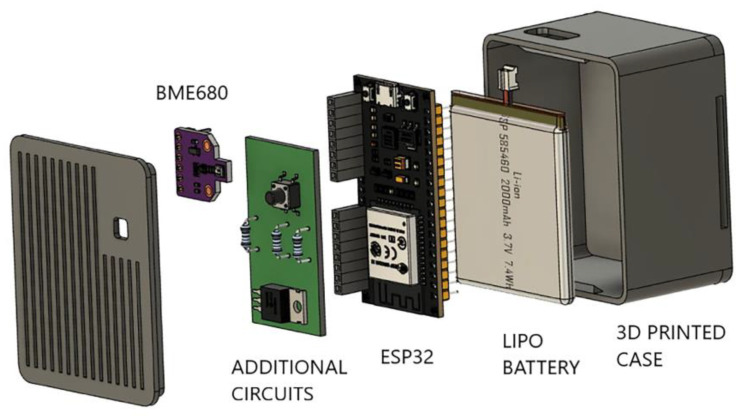
Schematic representation of the sensing node.

**Figure 4 sensors-23-02457-f004:**
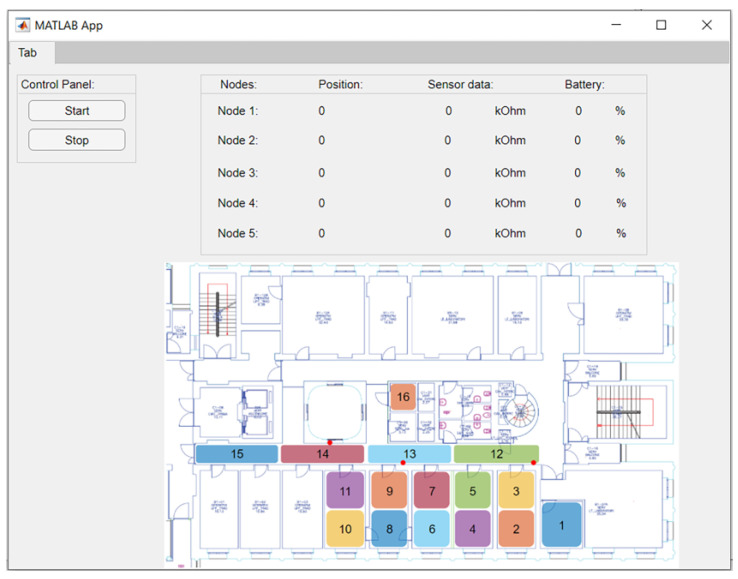
The MATLAB script GUI is designed for five sensing nodes enumerated from 1 to 5. When one of the five sensing nodes starts to advertise, the GUI shows position, area, sensor data, and battery level.

**Figure 5 sensors-23-02457-f005:**
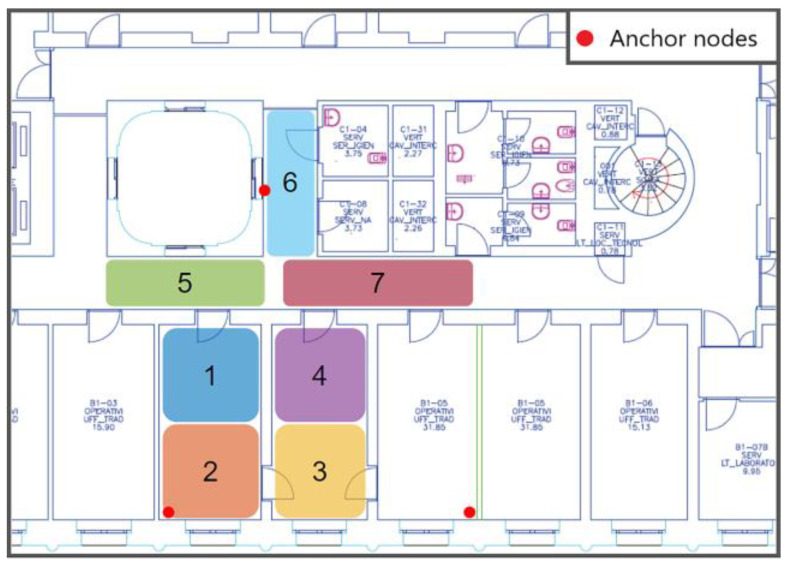
Test site map. The seven areas for the first experiment are highlighted. The red dots represent the anchor nodes.

**Figure 6 sensors-23-02457-f006:**
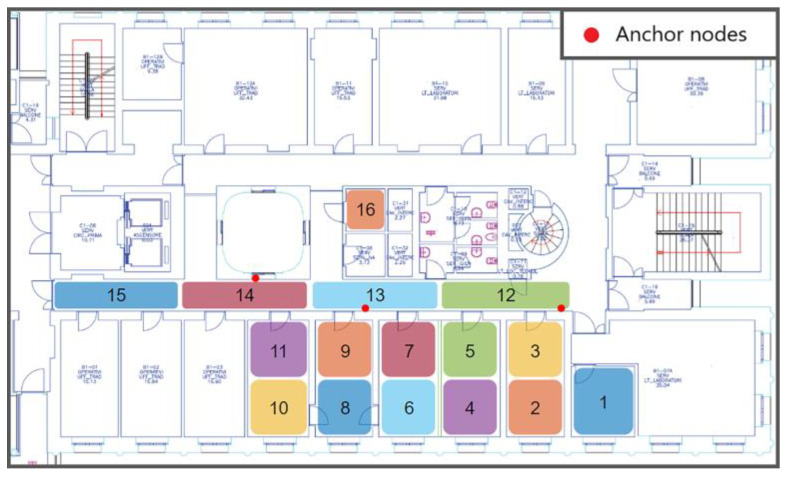
Test site map. The sixteen areas for the second experiment are highlighted. The red dots represent the anchor nodes.

**Figure 7 sensors-23-02457-f007:**
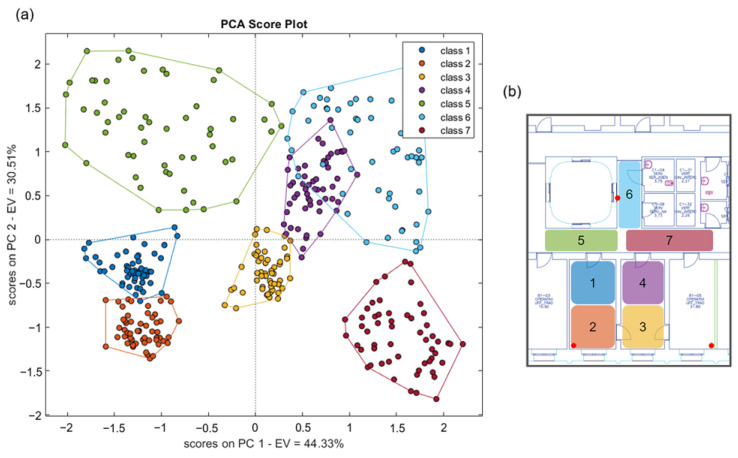
Scores plot of the first two principal components of the PCA model performed using the maximum value as a feature and a moving average filter of order six (**a**). The seven zones related to the seven classes are represented in (**b**).

**Figure 8 sensors-23-02457-f008:**
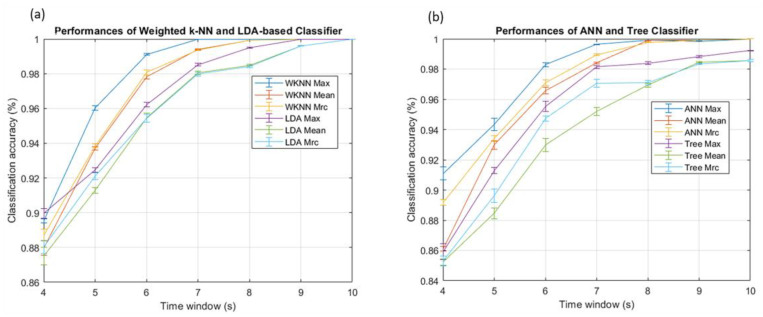
Classification performances of weighted k-NN, LDA-based classifier (**a**), ANN, and Tree Classifier (**b**) using a moving average filter.

**Figure 9 sensors-23-02457-f009:**
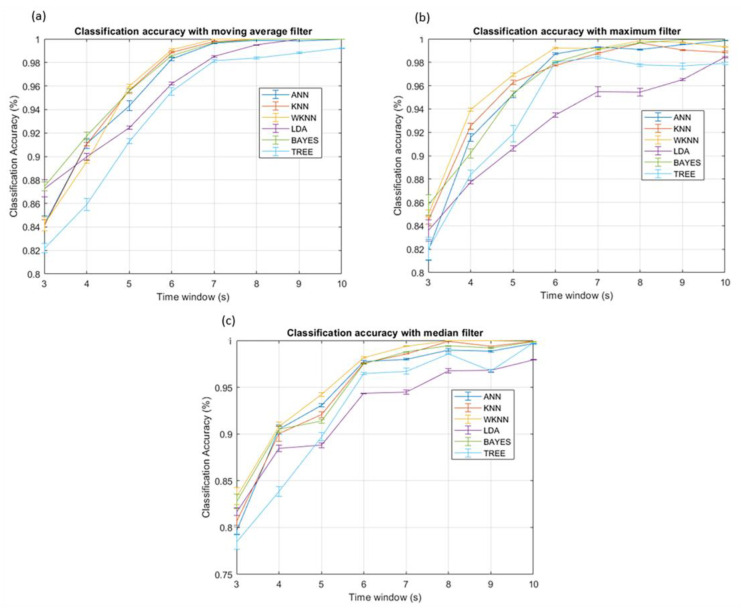
Performances of different classifiers when it is applied the moving average filter (**a**), the maximum filter (**b**), and the median filter (**c**).

**Figure 10 sensors-23-02457-f010:**
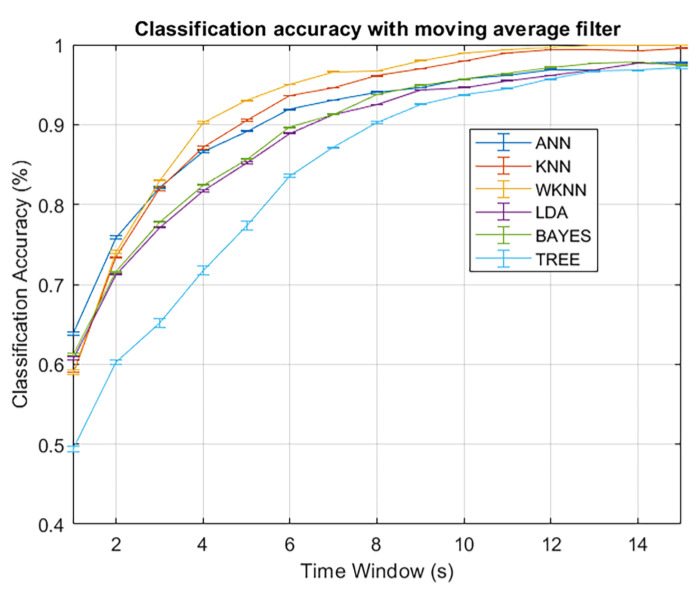
Performances of different classifiers in the second experiment using the moving average filter.

**Figure 11 sensors-23-02457-f011:**
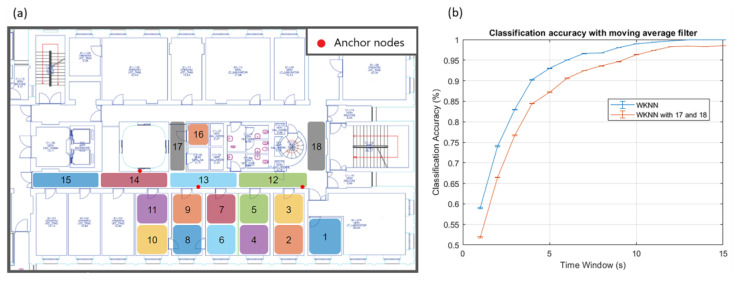
Test site map. The two additional areas (17 and 18) are gray (**a**). Drop-in performances are determined by the presence of the new regions (**b**).

**Figure 12 sensors-23-02457-f012:**
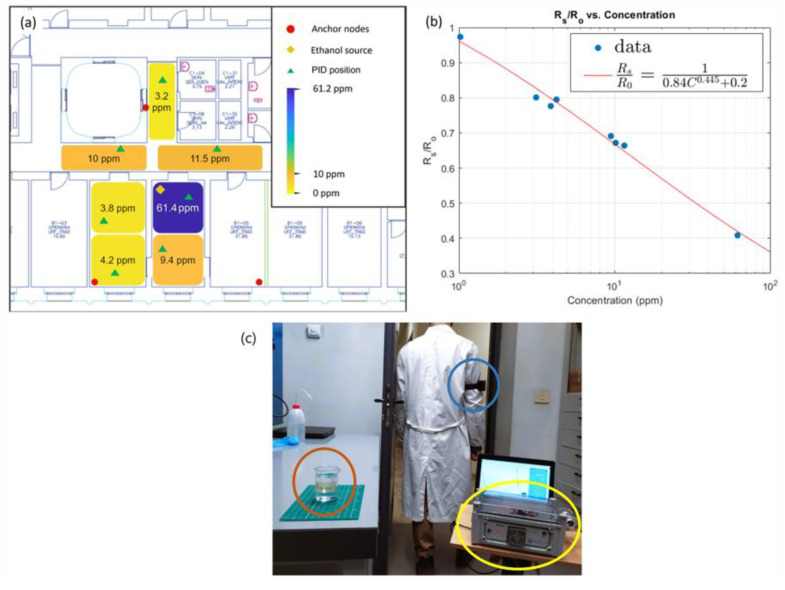
Ethanol source placement in the test site. Red dots show the location of the anchor nodes, green triangles are the PID fixed positions, and the yellow diamond is the ethanol source. We reported the concentration measured with PID in each zone (**a**). Relationship between the BME680 MOx resistance and the ethanol concentration measured with PID (**b**) Experimental setup: red circle shows the ethanol source, the yellow circle shows the PID, and the blue circle shows the sensing node worn by the operator (**c**).

## Data Availability

All data that support the findings of this study are available after the reasonable request to the corresponding author.

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
