# Peer review of "Personal VOCs Exposure with a Sensor Network Based on Low-Cost Gas Sensor, and Machine Learning Enabled Indoor Localization"

_sensors, 2023, doi:10.3390/s23052457_

Round 1

Reviewer 1 Report (New Reviewer)

The manuscript is interesting and relevant. It presented a WSN-based monitoring system that detects and processes VOCs’ information in an indoor area based on integrated gas sensors technology, BLE, communication, and machine learning algorithms. The paper demonstrated the depth of knowledge in effective WSN application and is scholarly written, organized, and presented logically and technically. The methods employed were appropriate and comprehensively discussed. The results obtained were clearly presented and the conclusion was appropriately drawn.

I, therefore, accept the paper as it is. However, the authors should proofread the paper thoroughly for typos and grammatical errors.

Author Response

Thanks for the helpful comment. We have checked and corrected the grammar and format of the paper.

Reviewer 2 Report (New Reviewer)

My review comments are in the attachment.

Author Response

In this paper, a monitoring system based on a Machine Learning approach was proposed to demonstrating the simultaneous detection and localization of the VOC source. I have some comments as follows:

  1. There are some formatting and grammar errors. For example, 'are an' appears in line 39 and 'm' is not in italics in line 243. Please correct the errors in grammar and format.

Thanks for the valuable comment. We have checked and corrected the grammar and format.

  1. The author uses RSSI for positioning, but the paper does not give a detailed introduction to the positioning of RSSI, such as the ranging principle of RSSI, ranging model and how to use RSSI for positioning, etc. Please supplement RSSI in more detail.

Thanks for the helpful remark. We have added a proper description in the introduction of RSSI-based positioning strategies.

  1. In my opinion, the chapter arrangement of the article is a little unreasonable. Some settings of the experiment are placed in "2. Materials and Methods", which should be included in "3. Results and Discussion". However, the characteristics and signal filtering of RSSI and classification model which should appear in "2. Materials and Methods" are put in "3. Results and Discussion". Please optimize the structure of the article.

Thank you for this remark.  We fixed the arrangement of paragraphs based on your suggestions.

  1. More experimental details should be given in this paper, such as the source of dataset which training classifier and the parameter settings of classifier, etc.

Thank you for this remark. The parameter settings of classifiers, as well as the description of how the training was performed, are reported in the paragraph “Classification models”. We added some missing information about k-NN models. The dataset is collected in the calibration process, 70 % of the dataset was used for the training and 30 % for the test.

  1. In this paper, VOC leakage experiment is an actual experiment or a simulation experiment. It would be better if there are some actual pictures of the test site attached to the actual experiment

Thank you for the comment. It is an actual experiment. We have added a picture of the experiment (Fig. 12 c)

Reviewer 3 Report (New Reviewer)

The manuscript titled "Personal VOCs exposure with a sensor network based on low-cost gas sensor, and machine learning enabled indoor localization" authored by Papale et. al., reports on the development of wireless network-based sensors for ethanol detection indoors. The authors also incorporated a machine learning approach to process the information delivered at a low cost by fabricating wearable volatile organic compound sensors. The WSN included fixed anchor nodes necessary for the localization of mobile devices. The localization of mobile devices is performed by analyzing the RSSIs with machine learning algorithms aimed at localizing the emitting source in a predefined map. The WSN was equipped with a commercial metal-oxide semiconductor gas sensor, to map the distribution of ethanol from a point-like source. The sensor signal is correlated with ethanol concentration measured by a PhotoIonization Detector (PID), demonstrating the simultaneous detection and localization of the VOC source.  While this was an interesting study of WSN-based VOC sensors for specific ethanol, there are some minor editorial and mechanical deficiencies. The authors are recommended to re-read the manuscript and correct it accordingly. Also, look into the Figures 8 and 12 captions and the contents. Once it is taken care of, the paper is acceptable for publication in this journal.

Author Response

Thanks for the helpful comment. We re-read the manuscript and corrected it.

Reviewer 4 Report (New Reviewer)

The author used WSN and adopted the method of machine learning and RSSI to locate and analyze VOCs exposure. This paper focuses on the signal processing and data analysis of RSSI. The authors compare multiple machine learning algorithms. WKNN exhibits the best efficiency. The authors used a realistic test environment to train the localization model. Accurate positioning of an area of 120m^2 is achieved with only three anchor notes.

Comments

1. Can the author explain in detail how the classification accuracy that appears multiple times in the text is defined and calculated?

2. The 8-hour exposure limit for most hazardous chemicals is at the ppb level. At this level, low-toxicity VOCs (plants, perfumes, air fresheners) in the environmental background will greatly affect the detection of hazardous substances. The detection of hazardous substances that cannot be qualitative and quantitative at the same time is not convincing. People usually use VOC Chek to absorb VOCS in the environment and use GC/MS for post-analysis.

3. Can the author explain the definition of i in ????i in eq1?

4. The author took into account the interference of the movement of people on the Bluetooth signal. But can you explain the influence of wifi, cellular network, and microwave oven on the positioning results?

5. Trilateration or fingerprinting is a simple RSSI data processing method that can be used for positioning. Can the authors further explain the advantages of machine learning over simpler methods?

Author Response

Thanks a lot for your helpful comments. We answered in the word attached.

This manuscript is a resubmission of an earlier submission. The following is a list of the peer review reports and author responses from that submission.

Round 1

Reviewer 1 Report

This paper presents a WSN-based indoor localization system with many possible applications, such as analysing workers’ exposure to chemicals. Different machine learning (ML) approaches were tested to evaluate the position from the RSSI triplets measured by the anchor nodes. Then, the performance of these ML algorithms was extracted and compared. The paper is well-written and organized. The topic is interesting in this field. However, the paper needs significant improvements before it should be accepted as follows:

  1. In the abstract, the localization percentage of 99% should be clearly stated this value is related to weighted k-NN.
  2. The introduction section does not provide sufficient background. Some previous articles related to k-NN, weighted k-NN, and ANN for indoor localization can be included and discussed.
  3. The pros and cons of using RSSI can be discussed in an introduction section. See: 10.1109/ACCESS.2020.3016832
  4. In WSN localization, the crucial factor is the localization error or localization accuracy. Therefore, the authors must evaluate one of them for all adopted machine learning algorithms.
  5. Figures 1 and 2 are not explained in the text. They should be explained.
  6. A Block diagram of the whole proposed system must be provided.
  7. The wireless communication technology between sensing, anchor, and central nodes must be defined. Is it Wi-Fi or BLE?
  8. Section 2.1 (WSN-based monitoring system) is not well explained. It must be rewritten.
  9. The authors should justify using a metal oxide (MOx) gas sensor.
  10. How did the author obtain the RSSI measurement from BLE, and which device they are using? It should be highlighted.
  11. In Figure 2, the BLE is not included.
  12. Complete the sentence in line 161, “but the communication system [13]”.
  13. The sentence in line 163, "a classical RSSI-based approach, often ensures excellent performances," is not correct; revised it. The classical RSSI-based approach is inaccurate due to RSSI fluctuation, especially in indoor environments.
  14. Based on which criteria the anchor nodes were distributed.
  15. In line 173, the 120 measurements are collected from one position of the sensor node position or different positions inside the interested area. It should be clarified.
  16. Figure 3 shows five nodes. The authors should clarify which one is the sensor, anchor, and central nodes.
  17. In line 187, "Assuming that the position does not change so much in this small interval," the experiment should be repeated for a movable node to see the effect of the indoor environment on the RSSI measurement, such as multipath fading, interferences, etc. which produced from the wall, furniture, windows, door, etc.
  18. The indicators in Sections 2.3, 2.4 and 2.5 should be numbered as equations.
  19. The hyperparameters of the ANN must be provided.
  20. The authors used 70% of the data for training and 30% for testing in the classification. It should be justified. What about the data set of the validation process.
  21. It is not clear how the classification rate was evaluated.
  22. In figure 10, the caption of Figure 10(b) is missing.
  23. VOCs should be defined for the first appearance. Also, check other abbreviations.
  24. Accessed time for references in the list of references must be included.

Reviewer 2 Report

The article describes the main advantages of using a customised indoor localisation system. Some important issues arise, which are described in the following points:

  • The title is misleading as the study focuses mainly on the indoor localisation system and the reference to VOCs is not justifiable.
  • Considering the previous point, the abstract could be revised. In addition, the acronym RSSI should be made explicit.
  • The "Introduction" section needs to be revised as it contains a lot of text describing how the system was designed. This part can be moved to the "Methodology" section. In the "Introduction" section you need to deal in detail with the state of the art, which can also be used to justify your work.
  • Line 47: Please use capital letters when introducing an acronym.
  • Figure 2 and all others: You must introduce all figures beforehand in the text
  • Line 239: please revise "circa".
  • Section 3, Results and Discussion (major issue): it needs to be revised because I do not understand why you start this section with two figures without including a short introduction. The results are not consistent enough to give the reader an impression of the quality of the proposed work

Reviewer 3 Report

Dear Authors

The manuscript presents an interesting solution that integrates a chemical substances (i.e. VOC) concentration monitoring system with an Bluetooth-based indoor location system. The novelty of the solution is not only the measurement of the spatial distribution of VOC concentrations, but the determination the total exposure to hazardous materials for people in motion.

The concept presented in the first part of the article is quite interesting, but later the authors focused only on the analysis of the location method. The paper presents the results of an experiment which is a variation of the well-known fingerprints-based localization method. The research was carried out correctly. The influence of the signal filtering method and the selection of classification methods on the positioning accuracy were tested and well presented. Unfortunately, the authors did not provide any results regarding the total exposure for VOC. The article would benefit from such research.

In my opinion, the research work should be expanded and supplemented if you want to publish the article. Additionally, proofreading of the language and the grammar style will be required

More:

Suggests to redefine the “classification rate” to “Classification Accuracy”. Please use the confusion matrix and define accuracy basing on its. You can also estimate the accuracy for every area separately. According to fig 6, they can be strongly different.

It’s worth to define approx. size of areas to estimate positioning accuracy in [m].

The formulas are not numbered

Consider to use exponential moving average (EMA, DMA) or Kaufmann’s filters to improve filtering of RSSI values.

Finally: Note that in order to determine the cumulative exposure to VOC, the location knowledge is not needed if the sensor is personal (worn by the workers). However, knowing the location is important if there are additional stationary sensors in the rooms.

Round 2

Reviewer 1 Report

The authors have given a detailed reply to the reviewers and have addressed all the comments. The paper can be accepted for publication in the current form.

Author Response

We thank the reviewer for the helpful comment.

Reviewer 2 Report

The new section 3.2. VOC experiments, introduced in response to my comment to justify the reference to VOCs in the title, is quite qualitative and in my opinion does not meet the requirements to justify publication in SENSORS - MDPI: I consider it more suitable for a conference proceeding. In general, the revised version with the changes highlighted in yellow, especially with the introduction of the new section 3.2, is not sufficient to justify the publication of the article in SENSORS Journal in its current form.

Author Response

We thank the reviewer for the comment. The experiment about ethanol distribution has been repeated, complementing the sensor response with the concentration measured with a PID.

Reviewer 3 Report

Thanks for submitting the improved and supplemented manuscipt. In my opinion this version is worth to publish.

  Minor: Line 433 - formula not numbered

Author Response

(The authors gave the same response as above.)

Round 3

Reviewer 2 Report

I have also seen the latest version, but the improvement is not sufficient for me to justify publishing the article in Sensors Journal.